# Mango-GS: Enhancing Spatio-Temporal Consistency in Dynamic Scenes Reconstruction using Multi-Frame Node-Guided 4D Gaussian Splatting

**Tingxuan Huang** [1], **Haowei Zhu** [1], **Jun-hai Yong** [1], **Hao Pan** [1], **Bin Wang** [1,2*]
[1] School of Software, Tsinghua University [2] BNRist
`{htx25, zhuhw23}@mails.tsinghua.edu.cn,`
`{yongjh, haopan, wangbins}@tsinghua.edu.cn`

## Abstract

Reconstructing dynamic 3D scenes with photorealistic detail and strong temporal coherence remains a significant challenge. Existing Gaussian splatting approaches for dynamic scene modeling often rely on per-frame optimization, which can overfit to instantaneous states instead of capturing underlying motion dynamics. To address this, we present Mango-GS, a multi-frame, node-guided framework for high-fidelity 4D reconstruction. Mango-GS leverages a temporal Transformer to model motion dependencies within a short window of frames, producing temporally consistent deformations. For efficiency, temporal modeling is confined to a sparse set of control nodes. Each node is represented by a decoupled canonical position and a latent code, providing a stable semantic anchor for motion propagation and preventing correspondence drift under large motion. Our framework is trained end-to-end, enhanced by an input masking strategy and two multi-frame losses to improve robustness. Extensive experiments demonstrate that Mango-GS achieves state-of-the-art reconstruction quality and real-time rendering speed, enabling high-fidelity reconstruction and interactive rendering of dynamic scenes.

## 1 Introduction

Reconstructing photorealistic and dynamic 3D scenes from casually captured videos is a significant goal in computer vision and graphics. Success in this area enables a wide range of applications, including virtual reality, digital twins, and advanced visual effects Tang et al. (2023); Wu et al. (2025b). Early methods for this task, particularly those based on Neural Radiance Fields (NeRF), have demonstrated compelling results in capturing scene appearance and geometry Attal et al. (2023); Li et al. (2022a); Park et al. (2021b); Pumarola et al. (2021); Wang et al. (2023). However, NeRF-based approaches such as D-NeRF, Nerfies, and HyperNeRF, often require extensive computational resources and suffer from slow rendering speeds. A more recent method, 3D Gaussian Splatting (3DGS), brought a major breakthrough for static scenes Kerbl et al. (2023). It achieves high quality rendering in real time by the direct rasterization of 3D Gaussians.

Inspired by its success, researchers have begun extending 3DGS to model dynamic scenes. Many current approaches learn the motion of Gaussians by introducing time-dependent parameters or using a deformation network, typically a Multi-Layer Perceptron (MLP), to predict changes for each frame Wu et al. (2024); Park et al. (2025). While effective for simple movements, we observe a fundamental limitation in this per-frame optimization strategy. By processing each frame in isolation, these models tend to memorize the specific state of the scene at each moment rather than learning the underlying principles of motion. This makes it difficult to maintain temporal consistency, often leading to visual artifacts or blur, especially when dealing with fast or complex movements.

A natural way to improve temporal consistency is to consider multiple frames at once, allowing the model to understand motion trends Blattmann et al. (2023). Sequence models like Transformers

---

*Corresponding author.

Vaswani et al. (2017) are exceptionally well suited for this, as they can capture complex relationships across a temporal window. However, a significant obstacle arises when applying this idea to 3DGS. A typical scene contains often millions of individual Gaussians. A straightforward application of a temporal Transformer to all Gaussians would lead to prohibitive computational and memory costs, directly contradicting the core advantage of 3DGS, which is its efficiency and speed.

To overcome this challenge, we can simplify the complex motion of the entire scene by modeling the movement of a sparse set of control points. This principle has been explored by methods like SC-GS Huang et al. (2024), which uses a k-Nearest Neighbors (k-NN) approach to propagate motion from control nodes to the full set of Gaussians. This is a promising direction, yet it faces its own difficulties. When motion becomes rapid, the spatial neighborhood defined by k-NN in the initial frame may no longer be meaningful. For example, points that start close together can end up far apart, causing unrelated parts of the scene to be incorrectly influenced by the same control points. This weakens the model's ability to reconstruct challenging regions of large movements.

Our work addresses these limitations through two core insights. First, to prevent neighborhood drift, we propose a decoupled control node representation. Instead of treating a node as just a 3D position, we separate its spatial location from a persistent latent feature code. This decoupling allows the model to establish semantic neighborhoods. Second, on top of this representation, we introduce a multi-frame temporal attention mechanism. We use a temporal Transformer to learn coherent, physically plausible motion patterns, rather than memorizing individual frames.

To sum up, we introduce Mango-GS: a Multi-frame Node-Guided Optimization framework for Gaussian Splatting. Our main contributions are:

1. We introduce a decoupled representation for control nodes, which separates their spatial position from a latent feature code. This design effectively stabilizes the motion influence throughout the sequence and prevents neighborhood drift in scenes with large movements.

2. We apply a multi-frame temporal Transformer to model the dynamics of control nodes for Gaussian Splatting. This allows our model to capture complex, long-range temporal dependencies and generate highly coherent motion.

3. The resulting framework, Mango-GS, sets a new state-of-the-art in dynamic scene reconstruction, demonstrating superior performance in rendering quality and speed, particularly in challenging scenes with fast and intricate motion.

## 2 RELATED WORKS

**Dynamic Neural Rendering.** Reconstructing dynamic 3D scenes is a fundamental problem in computer vision. Following the breakthrough of NeRF Mildenhall et al. (2021) for high-quality neural rendering, substantial effort has gone into adapting NeRF to dynamics. Some approaches Li et al. (2022b; 2021) attach time-conditioned latent codes to represent scene evolution, while another line of work Park et al. (2021a;b); Pumarola et al. (2021) introduces explicit deformation fields that warp rays into a canonical space where a static NeRF is optimized. Building on efficient NeRF variants Song et al. (2023); Müller et al. (2022), the other methods Cao & Johnson (2023); Fridovich-Keil et al. (2023); Shao et al. (2023) further factorize the 4D space–time domain into sets of 2D feature planes to reduce model size and speed up training. Despite these advances, their inference speed still falls short of practical, real-time rendering requirements.

**Dynamic 3D Gaussian Splatting.** 3D Gaussian Splatting (3DGS) enables real-time, high-fidelity rendering for static scenes and has motivated deformable extensions that animate a canonical Gaussian set over time. Representative methods include D-3DGS Luiten et al. (2024) and Deformable 3DGS Yang et al. (2024), which regress per-Gaussian translation/rotation/scale with lightweight MLPs. This paradigm has been strengthened by richer inputs, such as the learnable embeddings in E-D3DGS Bae et al. (2024) and DN-4DGS Lu et al. (2024a). Event-boosted Deformable 3DGS further combines event streams with deformable Gaussians to recover high-speed motion that is hard to capture with RGB-only inputs Xu et al. (2025). To improve temporal modeling, 4DGaussians Wu et al. (2024) augments Gaussians with spatio-temporal encodings, while GaGS Lu et al. (2024b) explicitly extracts 3D geometry features and injects them into the deformation learner to enforce geometric coherence. In parallel, MotionGS Zhu et al. (2024b) introduces flow-based motion guidance to decouple camera motion from object motion, and TimeFormer Jiang et al. (2025) applies

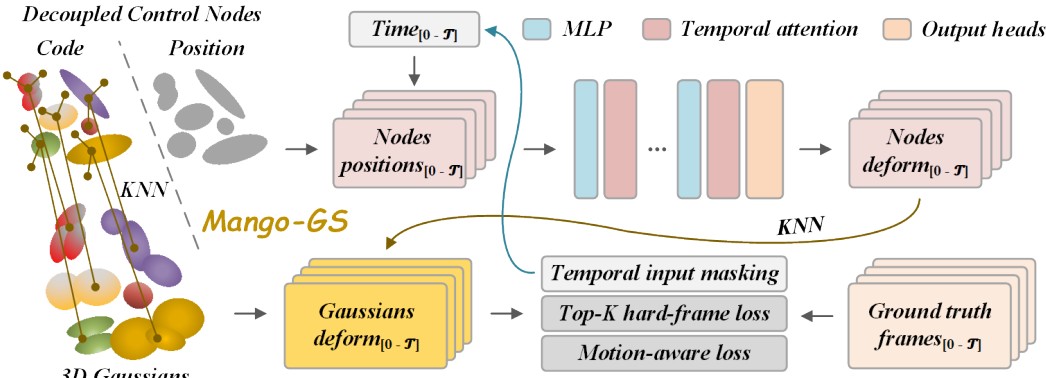

Figure 1: An overview of the Mango-GS framework. Our method is driven by a set of decoupled control nodes, each comprising a canonical position and a feature code. A dense 3D Gaussian cloud is associated with these nodes via a learned k-NN relationship based on both position and features. A temporal attention network takes the canonical node positions and a time window $[0, T]$ as input, processing them through MLP layers and temporal attention blocks to predict the nodes' deformations over the entire time window. This learned motion is then propagated back to the Gaussian cloud to produce the final dynamic scene representation for each frame. The entire model is optimized end-to-end with a temporal input masking scheme and a composite loss, which includes a top-$k$ hard-frame photometric loss, and a motion-aware temporal loss.

cross-temporal Transformers to deformable Gaussians, explicitly modeling temporal relationships between frames. For long sequences and efficiency, SWinGS Shaw et al. (2024) trains sliding temporal windows guided by 2D optical flow, and Swift4D Wu et al. (2025a) leverages a 4D hash grid to focus computation on dynamic regions.

Beyond per-Gaussian prediction, structure priors and explicit parameterizations have been explored. SC-GS Huang et al. (2024) uses a sparse set of control nodes to drive dense Gaussians via spatial k-NN. Our method follows this node-guided paradigm, but differs by (i) decoupling each node into a canonical position and a latent code, and (ii) performing multi-frame temporal modeling directly in the sparse node space rather than on dense Gaussians. Another family directly parameterizes time. Duan et al. (2024); Li et al. (2024); Yang et al. (2023) represent dynamics with true 4D Gaussians, increasing expressiveness at the expense of higher storage and training overhead. Fully explicit 4D Gaussian variants Lee et al. (2024) and decomposed 4D hash encodings such as Grid4D Xu et al. (2024) further improve fidelity and compactness, while MEGA Zhang et al. (2025) study memory-efficient color parameterizations and adaptive allocation of 3D versus 4D Gaussians. Beyond view synthesis, Splatter a Video Sun et al. (2024) represents generic videos with 3D Gaussians and associated motions to support downstream video tasks such as tracking and editing. Our work, Mango-GS adopts a sparse node–guided formulation with multi-frame modeling to enhance temporal coherence while retaining the efficiency benefits of Gaussian rasterization.

## 3 PRELIMINARY

**3D Gaussian Splatting.** 3DGS is an explicit point based representation in which each scene element is a 3D Gaussian parameterized by its mean $\mu \in \mathcal{R}^3$, positive definite covariance $\Sigma \in \mathcal{R}^{3 \times 3}$, per-point opacity $\alpha \in (0, 1]$, and view dependent color via spherical harmonics (SH). The Gaussian density is defined as:

$$G(x) = \exp\left(-\tfrac{1}{2}(x - \mu)^\top \Sigma^{-1}(x - \mu)\right). \tag{1}$$

To ensure $\Sigma \succ 0$, 3DGS factorizes it into a rotation $R \in SO(3)$ and a diagonal scaling $S = \mathrm{diag}(s)$ with $s \in \mathcal{R}^3_{>0}$: $\Sigma = R S S^\top R^\top$. For novel view synthesis, each 3D Gaussian is splatted onto the image plane. Let $W$ be the world to camera transform and $J$ the local affine or Jacobian at the projected mean. The covariance in image coordinates is $\Sigma' = J W \Sigma W^\top J^\top$, which defines a 2D Gaussian footprint used for rasterization. After projecting all Gaussians and sorting them along

the viewing ray from back to front, the per pixel color $C$ is computed by alpha compositing with effective opacities $\alpha_i'$ and SH shaded colors $c_i$:

$$C \;=\; \sum_{i=1}^{N} c_i\, \alpha_i' \prod_{j=1}^{i-1}\big(1-\alpha_j'\big), \tag{2}$$

where $c_i$ is evaluated from the Gaussian's SH coefficients under the current view direction, and $\alpha_i'$ is the splatted opacity derived from $\alpha$ and the projected footprint. The full pipeline is differentiable and supports end to end optimization of the Gaussian parameters for high-fidelity real time rendering.

## 4 METHOD

This section presents a high-level overview of the pipeline, followed by the two core components and the training strategy. We first summarize how sparse control signals are mapped to a dense, time-varying representation; we then describe the decoupled control node representation that stabilizes motion propagation; and we introduce the multi-frame temporal attention network that learns node dynamics; finally, we outline the overall optimization procedure. In addition, the structure of the training data is provided in Appendix A.1.

Our goal is to reconstruct dynamic 3D scenes from monocular or multi-view videos with both high-fidelity appearance and temporally coherent motion. To this end, we introduce Mango-GS (Fig. 3), which animates a dense 3D Gaussian representation using a sparse set of learnable control nodes. This sparse-to-dense design enables expressive temporal modeling with a compact attention network while avoiding the cost of operating directly on millions of Gaussians.

### 4.1 OVERALL PIPELINE

Our method represents a dynamic scene as a deformation of a canonical 3D Gaussian point cloud. The entire process, as depicted in Fig. 1, can be understood as a unified pipeline that maps a sparse set of control instructions to a dense, dynamic scene representation.

The foundation of our scene is a set of 3D Gaussians, $\mathcal{G} = \{g_j\}_{j=1}^{N}$, and corresponding sparse control nodes, $\mathcal{N} = \{n_i\}_{i=1}^{M}$, where $M \ll N$. Each control node $n_i = (p_i, f_i)$ is a decoupled entity composed of a canonical position $p_i \in \mathbb{R}^3$ and a learnable feature code $f_i \in \mathbb{R}^D$.

The influence of these control nodes on the dense Gaussian cloud is established through a learned k-Nearest Neighbor (k-NN) relationship. For each Gaussian $g_j$, we identify its $k$ most influential control nodes. The influence weight, $w_{ij}$, of node $n_i$ on Gaussian $g_j$ is computed based on their proximity in a joint position-feature space, ensuring a motion-aware neighborhood:

$$w_{ij} = \frac{\exp(-\mathcal{D}(g_j, n_i))}{\sum_{i' \in \mathcal{K}(j)} exp(-\mathcal{D}(g_j, n_{i'}))}, \quad \forall i \in \mathcal{K}(j), \tag{3}$$

where $\mathcal{K}(j)$ is the set of $k$ nearest nodes to Gaussian $g_j$, and $\mathcal{D}$ measures distance in a joint space of Gaussian canonical parameters and node features.

To animate the scene, we model the trajectory of all control nodes over a time window of $T$ frames. As shown in the central part of Fig. 1, our temporal attention network $\Phi$ takes the canonical node positions $p_i$ (tiled across the $T$ timestamps) and timestamps $t_0, \ldots, t_{T-1}$ as input. The network then predicts a sequence of deformations for each node all at once:

$$\{\Delta p_i(t),\, \Delta q_i(t),\, \Delta s_i(t)\}_{t=0}^{T-1} = \Phi(\{p_i\}_{i=1}^{M},\, \{t\}_{t=0}^{T-1};\, \Theta), \tag{4}$$

where $\Delta p$, $\Delta q$, and $\Delta s$ represent the predicted translation, rotation, and scaling for each node, respectively, and $\Theta$ are the learnable parameters of the network.

Finally, the node deformations are propagated to all Gaussians using the pre-computed k-NN weights. The deformation of each Gaussian $g_j$ at time $t$ is a weighted blend of its semantic neighboring control nodes. For instance, its positional deformation $\Delta(x_j)(t)$ is calculated as:

$$\{\Delta(x_j)(t)\}_{t=0}^{T-1} = \sum_{i \in \mathcal{K}(j)} w_{ij} \times \{\Delta p_i(t)\}_{t=0}^{T-1}. \tag{5}$$

Similar blending is applied to compute the final rotation and scaling for each Gaussian. The resulting deformed Gaussian cloud $g_j(t)$ can then be rendered from any viewpoint. The entire framework is trained end-to-end with a temporal input masking scheme and a composite loss function that evaluates the quality of rendered frame groups against ground truth data, ensuring photometric accuracy, robustness to challenging frames, and high temporal fidelity.

## 4.2 DECOUPLED CONTROL NODE REPRESENTATION

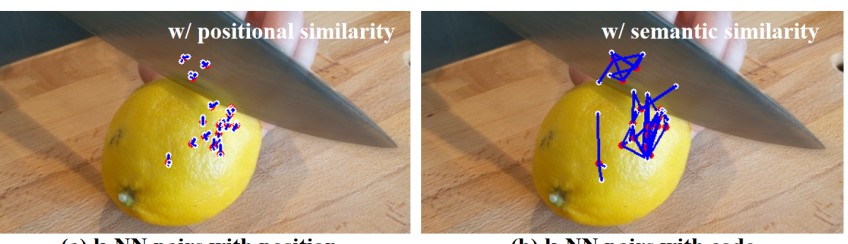

(a) k-NN pairs with position        (b) k-NN pairs with code

Figure 2: Position-only nodes versus decoupled nodes with position and code. We visualize 24 Gaussians (red) in high motion region and its three corresponding nodes (white). With the decoupled design, Gaussians attach to semantically consistent nodes rather than merely following spatial neighbors, which struggle under large motion.

A central challenge in modeling dynamic scenes lies in establishing a robust and meaningful correspondence between the sparse nodes and the dense Gaussian cloud. Purely using spatial k-NN in canonical space, as in SC-GS Huang et al. (2024), is fragile. Specifically, under large non-rigid motion, points that start nearby can end up on different, independently moving parts of the same or different objects. Relying on a static spatial neighborhood can therefore lead to erroneous motion propagation, causing blurring deformations and visual artifacts.

We address this by decoupling each control node into a position and a code, and by learning affinity rather than relying on Euclidean distance alone. The relationship is inferred from the full canonical parameters of each Gaussian, yielding neighborhoods that reflect shape, orientation, and location rather than proximity only.

Formally, let a Gaussian $g_j$ have canonical parameters $\phi_j = (x_j, q_j, s_j)$, where $x_j$, $q_j$, and $s_j$ represent its canonical position, rotation, and scaling factors, respectively. We compute a learned affinity score between $g_j$ and node $n_i$ using a lightweight MLP, and convert the top-$k$ scores into weights via softmax. The MLP maps Gaussian canonical parameters to an embedding that is compared with the node code to produce the affinity score. The final influence weights $w_{ij}$ for the $k$ most influential nodes are then derived by applying a softmax function to these scores.

This design brings two benefits. First, the learned affinity produces semantics-aware and robust neighborhoods, as evidenced in Fig. 2 (b), the longer KNN links indicate non-local yet consistent matches that remain valid under large displacement, unlike the short, purely local links in Fig. 2 (a). Second, it is parameter efficient by reusing existing Gaussian attributes, avoiding per-Gaussian latent features and the associated memory overhead.

## 4.3 TEMPORAL ATTENTION FOR NODE DYNAMICS

To generate realistic and temporally coherent motion, the model has to reason across time rather than memorize frame-wise states. Per-frame deformation, which treats frames in isolation, often fails to

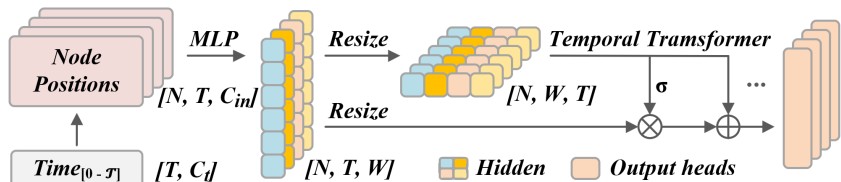

Figure 3: The Architecture of the Temporal Deformation Network. For each of the $N$ control nodes, features over a window of $T$ frames are processed by an MLP backbone interleaved with temporal self-attention blocks. Attention operates along the time axis and is fused by a lightweight gate, then decoded to per-node translation, rotation, and scale for all $T$ frames.

capture complex trajectories. We therefore design a multi-frame deformation network $\phi$ that models the dynamics of the control nodes across a time window of $T$ frames. As illustrated in Fig. 3, a deep MLP architecture is interleaved with temporal self-attention blocks that operate along the time axis, enabling the network to capture long-range, non-linear dependencies and produce coherent motion for all nodes within the window.

The network processes all $N$ control nodes simultaneously. For each node, its canonical position $p_i$ is first encoded into a feature $x_{emb}$. Similarly, timestamps $t_0, ..., t_{T-1}$ are embedded to $t_{emb} \in \mathcal{R}^{T \times C_t}$. These are concatenated and shaped into an initial feature tensor $H^{(0)} \in \mathcal{R}^{N \times T \times C_{in}}$, which summarizes all nodes over the window. $C_{in}$ is the input feature dimension. $H^{(0)}$ is then processed through a series of $L$ layers with the hidden size of $W$. Most layers are standard MLP blocks with ReLU activations. However, at specific layers $l$ within the network, we introduce the temporal attention block.

Given the input features $H^{(l)} \in \mathcal{R}^{N \times T \times W}$, the block first projects them into a temporal attention space $H_i n^{(l)} \in \mathcal{R}^{N \times W \times T}$. We then apply multi-head self-attention (MHA) along the temporal axis. For each node, this operation computes with all time steps within the window $[0, T-1]$:

$$H_{attn} = MHA(H_{in}^{(l)}, H_{in}^{(l)}, H_{in}^{(l)}). \tag{6}$$

We then use a lightweight gating module Chen et al. (2023); Huang et al. (2025) to generate $(w_{\text{gate}}, w_{\text{bias}})$ from $H_{\text{attn}}$ and fuse temporal information back into the main stream:

$$H(l+1) = H(l) \otimes (\sigma(w_{gate})) + w_{bias}. \tag{7}$$

This allows the network to adaptively control how much of the temporal information is incorporated. Here, $\sigma$ is the sigmoid function, $w_{gate} \in \mathcal{R}^{N \times W \times T}$ and $w_{bias} \in \mathcal{R}^{N \times W \times T}$ are output attention signals, and $\otimes$ denotes element-wise multiplication. This gated update rule is more expressive than a simple residual connection. After the final layer, the output tensor $H^{(L)}$ is passed through separate linear heads to decode the final deformations for each node at each of the $T$ time steps.

## 4.4 TRAINING AND OPTIMIZATION STRATEGY

The Mango-GS framework is trained end-to-end with two key strategies: temporal input masking scheme to strengthen motion reasoning, and a composite loss function that focuses on difficult frames while enforcing cross-frame consistency.

**Temporal input masking.** To prevent the temporal attention network from trivially memorizing timestamps and frame-specific appearance, we employ an input masking strategy. This can be viewed as a lightweight form of training-time augmentation that improves robustness Zhu et al. (2024a). During training, we randomly select and mask out a subset of the input time embeddings. The network is then tasked with predicting the deformations for the entire window using only the visible temporal context.

The overall loss function is:

$$\mathcal{L} = 0.8 \times \mathcal{L}_{\text{frame}} + 0.2 \times \mathcal{L}_{\text{motion}}, \tag{8}$$

where $\mathcal{L}_{\text{frame}}$ is a top-$k$ hard-frame photometric loss, and $\mathcal{L}_{\text{motion}}$ is a motion-aware loss.

**Top-$k$ hard-frame loss.** It is the primary supervision signal, which is based on photometric loss, calculated on a per-frame basis as a weighted combination of the L1 loss and DSSIM. Instead of averaging this loss across all frames, we apply a top-$k$ hard-frame mining strategy. The final image reconstruction loss, $\mathcal{L}_{\text{frame}}$, is the mean of the individual loss from only the k frames with the highest error in the batch. We set K to a fixed ratio of $0.6 \times$ batch size and recompute the top-$k$ frames at every iteration, so gradients are reweighted toward the currently hard frames. This focuses optimization on the most challenging moments in the sequence.

**Motion-aware loss.** To explicitly supervise temporal coherence, the model uses a motion-aware loss, $\mathcal{L}_{\text{motion}}$, that operates on the temporal differences. Let $\delta\hat{I}_t = \hat{I}_t - \hat{I}_{t-1}$ and $\delta I_t = I_t - I_{t-1}$ denote the temporal differences between consecutive rendered and ground-truth frames, respectively. We treat these differences as a compact description of how the scene evolves between neighboring timestamps. The motion-aware loss is defined as a weighted sum of three terms:

$$\mathcal{L}_{\text{motion}} = \lambda_{\text{diff}} \, \mathcal{L}_{\text{diff}} + \lambda_{\text{amp}} \, \mathcal{L}_{\text{amp}} + \lambda_{\text{dir}} \, \mathcal{L}_{\text{dir}}. \tag{9}$$

The three components are given by

$$\mathcal{L}_{\text{diff}} = \sum_{t,x} \left\| \delta\hat{I}_t(x) - \delta I_t(x) \right\|_1, \tag{10}$$

$$\mathcal{L}_{\text{amp}} = \sum_{t,x} \max\left(0, \ \|\delta I_t(x)\|_1 - \|\delta\hat{I}_t(x)\|_1\right), \tag{11}$$

$$\mathcal{L}_{\text{dir}} = \sum_{t,x} \left(1 - \cos(\delta\hat{I}_t(x), \ \delta I_t(x))\right), \tag{12}$$

Here $\cos(\cdot, \cdot)$ denotes the cosine similarity between temporal gradients (i.e., the dot product of $\ell_2$-normalized vectors), and we set $(\lambda_{\text{diff}}, \lambda_{\text{amp}}, \lambda_{\text{dir}}) = (0.7, 0.2, 0.1)$ in all experiments.

The first term, $\mathcal{L}_{\text{diff}}$, encourages the rendered sequence to exhibit similar frame changes as the real sequence, so that events such as object motion or appearance/disappearance occur with the correct spatial support. The second term, $\mathcal{L}_{\text{amp}}$ prevents the network from underestimating motion magnitude: even if the direction is correct, overly small temporal differences would lead to over-smoothed dynamics. The third term, $\mathcal{L}_{\text{dir}}$, focuses on the direction of change and penalizes cases where the rendered motion points in a different direction from the ground truth. Together, these terms ask the model not only to reconstruct individual frames, but also to match how they change over time, which substantially reduces flickering and yields more coherent motion.

## 5 EXPERIMENTS

### 5.1 EXPERIMENTS SETTINGS

**Datasets.** We evaluate our method on two representative real-world dynamic scene datasets: 1) HyperNeRF dataset Park et al. (2021b) is captured with 1-2 cameras, following straightforward camera motion. It contains complex dynamic variations, such as human movements and object deformations. 2) Neural 3D Video dataset Li et al. (2022b) contains 6 real-world scenes. These scenes feature relatively long durations and diverse motions, some containing multiple moving objects. Each scene has approximately 20 synchronized videos. Except for the flame salmon scene, which consists of 1200 frames, all other scenes comprise 300 frames. For each scene, we select one camera view for testing while using the remaining views for training.

**Metrics.** To evaluate reconstruction quality, we employ peak-signal-to-noise ratio (PSNR), structural similarity index (SSIM), multi-scale SSIM (MS-SSIM), and perceptual quality measure LPIPS with an AlexNet Backbone to assess the rendered images. Additionally, we evaluate storage efficiency by calculating the output file size as storage (MB). We also measured the rendering speed (FPS). On HyperNeRF dataset, we further report a temporal LPIPS (tLPIPS) Chu et al. (2018) metric computed on frame-to-frame differences to quantify temporal perceptual quality and temporal coherence.

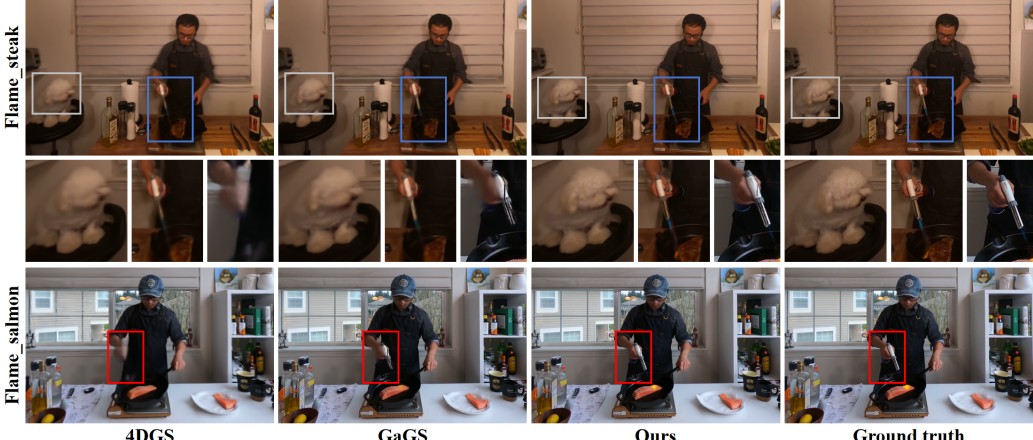

Figure 4: Visualization comparison between baselines and our methods from the Neural 3D Video dataset. The main differences are highlighted and zoomed in with boxes.

**Implementation details.** We adhere to the settings of the original 3DGS paper for the static attributes of our Gaussian representation. Our dynamic model operates on a time window of $T = 6$ frames and utilizes 2048 control nodes at the beginning to guide the scene deformation. The training process follows a two-stage optimization: a node initialization and warm-up stage spanning 5,000 iterations, followed by a main dynamic training stage of 35,000 iterations where all parameters, including the temporal attention network and the canonical Gaussians, are jointly optimized. For our composite loss function, the DSSIM weight $\lambda_{dssim}$ is set to 0.2, and the weights for our motion-aware loss components are determined through preliminary experiments. All experiments are conducted on a single NVIDIA RTX 3090 GPU.

## 5.2 COMPARISON WITH STATE-OF-THE-ART

Table 1: Quantitative comparison on two datasets. Best is **bold** and second-best is underlined.

| Method | Neural 3D Video | | | HyperNeRF-vrig | | | | |
|---|---|---|---|---|---|---|---|---|
| | PSNR↑ | SSIM↑ | LPIPS↓ | PSNR↑ | MS-SSIM↑ | tLPIPS↓ | FPS↑ | Storage↓ |
| D-3DGS | 31.15 | 0.941 | 0.078 | 25.0 | 0.70 | 0.0234 | 14.2 | 172 MB |
| E-D3DGS | 30.86 | 0.938 | **0.048** | 25.4 | 0.70 | 0.0257 | 45.2 | 64 MB |
| 4DGS | 31.58 | 0.942 | 0.055 | 25.2 | 0.68 | 0.0248 | 45.0 | **59 MB** |
| SC-GS | 30.20 | 0.935 | 0.067 | 23.6 | 0.66 | 0.0236 | 24.5 | 85 MB |
| GaGS | 31.10 | **0.944** | 0.060 | 24.3 | 0.65 | 0.0233 | 12.0 | 48 MB |
| MotionGS | - | - | - | 24.6 | 0.71 | 0.0229 | 39.9 | 69 MB |
| TimeFormer | 31.84 | 0.941 | - | 24.3 | 0.68 | 0.0265 | 40.9 | **46 MB** |
| **Ours** | **31.89** | 0.942 | 0.049 | **26.2** | **0.78** | **0.0196** | **149.5** | 60 MB |

We compare Mango-GS with recent dynamic-scene methods, including Deformable 3DGS (D-3DGS) Yang et al. (2024), E-D3DGS Bae et al. (2024), 4DGS Wu et al. (2024), SC-GS Huang et al. (2024), GaGS Lu et al. (2024b), MotionGS Zhu et al. (2024b) and TimeFormer (w/ 4DGS) Jiang et al. (2025). Unless otherwise noted, numbers are averaged over all test views of each dataset.

**Neural 3D video.** Mango-GS achieves the highest PSNR and competitive SSIM and LPIPS, as shown in Tab. 1. Deformable 3DGS reaches the second-best PSNR and SSIM but with higher LPIPS, and E-D3DGS improves perceptual quality at the cost of PSNR. 4D-GS and GaGS are competitive but fall short overall. We note that MotionGS do not report metrics on the Neural 3D Video benchmark in Tab. 1, so we primarily compare against the others. For visual assessment, qualitative comparisons in Fig. 4 show that our method produces superior results; as highlighted in

the boxed areas, Mango-GS renders sharper details in both the dynamic foreground subjects and the static background.

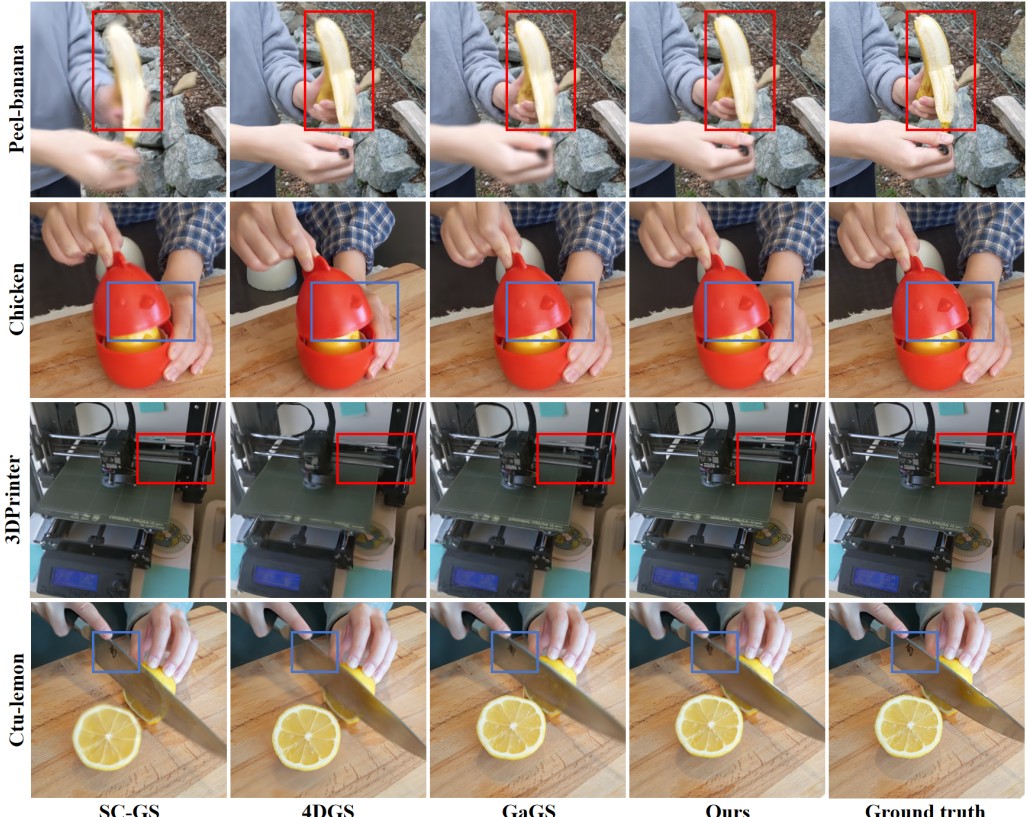

Figure 5: Qualitative comparison on HyperNeRf. Our method offers sharp results. Differences are highlighted with boxes.

**HyperNeRF.** Mango-GS achieves the best accuracy on HyperNeRF, as shown in Tab. 1. Our method attains the highest PSNR and MS-SSIM as well as the lowest temporal LPIPS (tLPIPS), outperforming strong baselines such as MotionGS and TimeFormer in both spatial and temporal quality. Thanks to our multi-frame inference framework, Mango-GS runs at 149.5 FPS, well above the other methods, while keeping its storage size at 60 MB, which is lower than Deformable 3DGS and SC-GS, and comparable to the others. In particular, Mango-GS is over $3\times$ faster than MotionGS and TimeFormer while achieving superior tLPIPS and MS-SSIM, indicating a better trade-off between efficiency and reconstruction fidelity. Furthermore, a per-scene breakdown on the HyperNeRF-vrig sequences is provided in the supplementary material (Tab. 6). To further assess performance, we provide qualitative comparisons in Fig. 5. Existing methods often struggle with fast or intricate motion, introducing noticeable blurriness and ghosting artifacts. In contrast, Mango-GS generates sharp and temporally coherent results, accurately capturing fine details in both static and dynamic regions.

## 5.3 ABLATION STUDIES

To validate our design choices and analyze the impact of key hyperparameters, we conduct a series of comprehensive ablation studies on the HyperNeRF dataset. We investigate both the parameters of our model and the contributions of its core architectural components.

**Time window and K-neighbors.** We study the impact of the temporal window size $T$ and the number of nearest neighbors $K$. The results are summarized in Tab. 2. For the time window $T$, we observe that performance peaks at $T = 6$. A smaller window is insufficient to capture long-range motion dependencies, while a larger window introduces a slight performance drop and significantly

Table 2: Ablation on time window size $T$ and number of neighbors $K$.

| | (a) Temporal Window Size ($T$) | | | | | (b) Number of Neighbors ($K$) | | |
|---|---|---|---|---|---|---|---|---|
| $T$ | PSNR ↑ | SSIM ↑ | tLPIPS ↓ | FPS ↑ | $K$ | PSNR ↑ | SSIM ↑ | tLPIPS ↓ |
| 2 | 27.53 | 0.925 | 0.0225 | 87.9 | 2 | 27.41 | 0.920 | 0.0205 |
| 4 | 28.19 | 0.937 | 0.0203 | 117.8 | **3** | **28.39** | **0.942** | **0.0196** |
| **6** | **28.35** | 0.942 | **0.0196** | 149.5 | 4 | 28.26 | 0.938 | 0.0199 |
| 8 | 28.24 | **0.940** | 0.0197 | **156.2** | 5 | 27.90 | 0.931 | 0.0203 |

Table 3: Ablation on the core components of Mango-GS. We report results from **progressively adding each component** to a baseline.

| Step | Method | PSNR ↑ | SSIM ↑ | LPIPS ↓ | tLPIPS ↓ |
|---|---|---|---|---|---|
| 1 | Baseline (single frame) | 25.15 | 0.875 | 0.139 | 0.0250 |
| 2 | + Nodes (w/o learned affinity) | 24.52 | 0.868 | 0.142 | 0.0235 |
| 3 | + Decoupled nodes (with learned affinity) | 25.31 | 0.892 | 0.118 | 0.0223 |
| 4 | + Multi-frame (w/o temporal attention) | 27.30 | 0.928 | 0.096 | 0.0225 |
| 5 | + Multi-frame (with temporal attention) | 27.78 | 0.937 | 0.084 | 0.0196 |
| 6 | + Top-$k$ loss | 28.05 | 0.941 | 0.077 | 0.0202 |
| 7 | + Motion-aware loss | 28.32 | 0.942 | 0.071 | 0.0192 |

reduces rendering speed, as it increases the sequence length for the attention mechanism. Our default setting of $T = 6$ provides the best balance of quality and performance, achieving 149.5 FPS. For the number of neighbors $K$, $K = 3$ yields the best results. A smaller $K$ may not provide enough information for stable deformation, while a larger $K$ tends to over-smooth the motion, losing fine-grained details. Consistent with these trends, tLPIPS improves as $T$ increases from 2 to 6 but slightly degrades at $T = 8$, suggesting that overly large temporal windows over-smooth the motion and hurt temporal perceptual quality; similarly, tLPIPS is minimized at $K = 3$, while larger $K$ values again lead to over smoothing and worse temporal coherence.

**Analysis of model components.** As shown in Tab. 3, starting from a single frame baseline, introducing nodes without learned affinity harms stability due to purely spatial propagation. Decoupling node position and code with a learned affinity restores reliable correspondences and improves detail. Moving to a multi-frame setting with a simple temporal MLP layers provides a clear boost by leveraging temporal context, and inserting temporal self-attention further benefits the performance. Adding the top-$k$ hard-frame loss sharpens difficult regions and improves perceptual quality. Finally, the motion-aware loss yields the most coherent temporal transitions, completing the full model. Numerically, tLPIPS exhibits clear drops when introducing decoupled nodes, temporal attention in the multi-frame setting, and the motion-aware loss, indicating progressively better temporal coherence. Although adding the top-$k$ loss alone may slightly increase tLPIPS while noticeably improving PSNR, combining it with the motion-aware loss yields the lowest tLPIPS, showing that focusing on hard frames and enforcing global motion consistency are complementary and jointly improve both spatial quality and temporal stability.

# 6 CONCLUSION

We introduced Mango-GS, a node-guided approach to dynamic scene reconstruction with Gaussian splatting. By decoupling control nodes into position and latent code, the method forms stable, semantics-aware neighborhoods; a multi-frame temporal attention module predicts coherent node trajectories that propagate to the dense Gaussians. The resulting system improves visual quality and temporal stability while maintaining high rendering speed. Future work includes extending the temporal horizon with lightweight memory, adapting online to long videos, and integrating stronger geometry or flow priors for extreme motions.

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

# A  APPENDIX

## A.1  MULTI-FRAME GROUP SAMPLING FOR TEMPORAL LEARNING

The effectiveness of the temporal attention network depends heavily on the quality and diversity of the training data. Thus we introduce two multi-frame group sampling strategies designed to teach the model about diverse dynamics and the continuous nature of time.

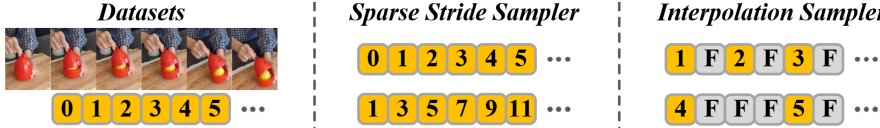

Figure 6: Overview of multi-frame group sampling strategies. Colored blocks with number denote real frames, and gray blocks with $F$ indicate placeholder timestamps. A row of blocks represents the $T$-frame training window.

**Sparse Stride Sampler.** From a view-specific sequence $\{(I_t, t)\}$, we pick $T$ real frames with a fixed stride. This approach creates training groups that span a longer duration, encouraging the network to learn a more robust understanding of dynamics.

**Interpolation Sampler.** To train our model to understand continuous motion rather than discrete states, this strategy mixes real frames with placeholder timestamps $\{\tilde{t}\}$ obtained by linear interpolation between nearby real times. Placeholders pass through the network but without any supervision. They act as temporal anchors to regularize the temporal attention, encouraging the model to learn a smooth function of time.

We slide the $T$-frame window over each timeline and alternate the two samplers with a fixed ratio (7:3), achieving long-range temporal coverage and local smoothness without increasing runtime.

Table 4: Ablation on the ratio between sparse stride and interpolation samplers.

| Sparse ratio | Interpolation ratio | PSNR↑ | MS-SSIM↑ | tLPIPS↓ |
|---|---|---|---|---|
| 1.0 | 0.0 | 25.4 | 0.74 | 0.0206 |
| 0.9 | 0.1 | 25.9 | 0.76 | 0.0199 |
| **0.7** | **0.3** | **26.2** | **0.78** | **0.0196** |
| 0.5 | 0.5 | 25.6 | 0.72 | 0.0210 |

We further study how the ratio between sparse stride and interpolation samplers affects learning. Intuitively, the sparse stride sampler exposes the network to larger temporal offsets and dynamics, while the interpolation sampler focuses on intermediate timestamps and encourages locally smooth motion. As shown in Table 4, PSNR and MS-SSIM remain relatively stable when the sparse sampler dominates, and the 0.7:0.3 setting used in Mango-GS achieves the best trade-off, giving the lowest tLPIPS. When the interpolation sampler dominates, both reconstruction quality and temporal consistency degrade, suggesting that strong long-range supervision is important for robust temporal modeling.

## A.2  ABLATION OF THE CONTROL NODES NUMBER

**Number of Control Nodes.** We analyze the effect of the initial number of control nodes, $N$. The nodes are dynamically densified and pruned during training, so we report both the initial and final counts. As shown in Tab. 5, performance improves as the number of nodes increases from 512 to 2048, as more nodes provide greater capacity to model complex deformations. However, further increasing the count to 4096 leads to a slight decline in performance, likely due to overfitting on the training views. Based on this, we use 2048 initial nodes as our default setting.

Table 5: Ablation on the number of control nodes $N$. We report initial and final node counts, along with reconstruction quality metrics.

| Initial Nodes | Final Nodes | PSNR ↑ | SSIM ↑ | LPIPS ↓ |
|---|---|---|---|---|
| 512 | ∼710 | 27.81 | 0.931 | 0.088 |
| 1024 | ∼960 | 28.14 | 0.938 | 0.079 |
| **2048** | **∼1420** | **28.32** | **0.942** | **0.071** |
| 4096 | ∼2550 | 28.25 | 0.941 | 0.073 |

Table 6: Per-scene reconstruction quality of Mango-GS on the HyperNeRF-vrig dataset

| Scene | PSNR↑ | MS-SSIM↑ | tLPIPS↓ |
|---|---|---|---|
| Broom | 23.3 | 0.72 | 0.0253 |
| Peel-Banana | 27.1 | 0.76 | 0.0213 |
| 3DPrinter | 26.3 | 0.79 | 0.0178 |
| Chicken | 27.7 | 0.85 | 0.0139 |
| Mean | 26.2 | 0.78 | 0.0196 |

## A.3 PER-SCENE RESULTS ON HYPERNERF-VRIG

As shown in Tab. 6, Mango-GS maintains consistently strong performance across all HyperNeRF-vrig scenes, and the per-scene scores aggregate to the same Mean values reported in the main comparison table.

## A.4 CONSECUTIVE QUALITATIVE RESULTS

Finally, to visually demonstrate the temporal coherence of our method, we present uncurated, consecutive rendering results in Figure 7. The sequences exhibit exceptional continuity, with seamless and natural transitions from one moment to the next. Our method successfully avoids flickering artifacts and maintains sharp, clear object boundaries. This high level of temporal stability is a direct result of our temporal attention network, which learns a continuous representation of motion rather than memorizing discrete frames. Furthermore, these high-fidelity results are rendered in real time, achieving 155.6 FPS at a resolution of $544 \times 960$, underscoring the efficiency of our framework.

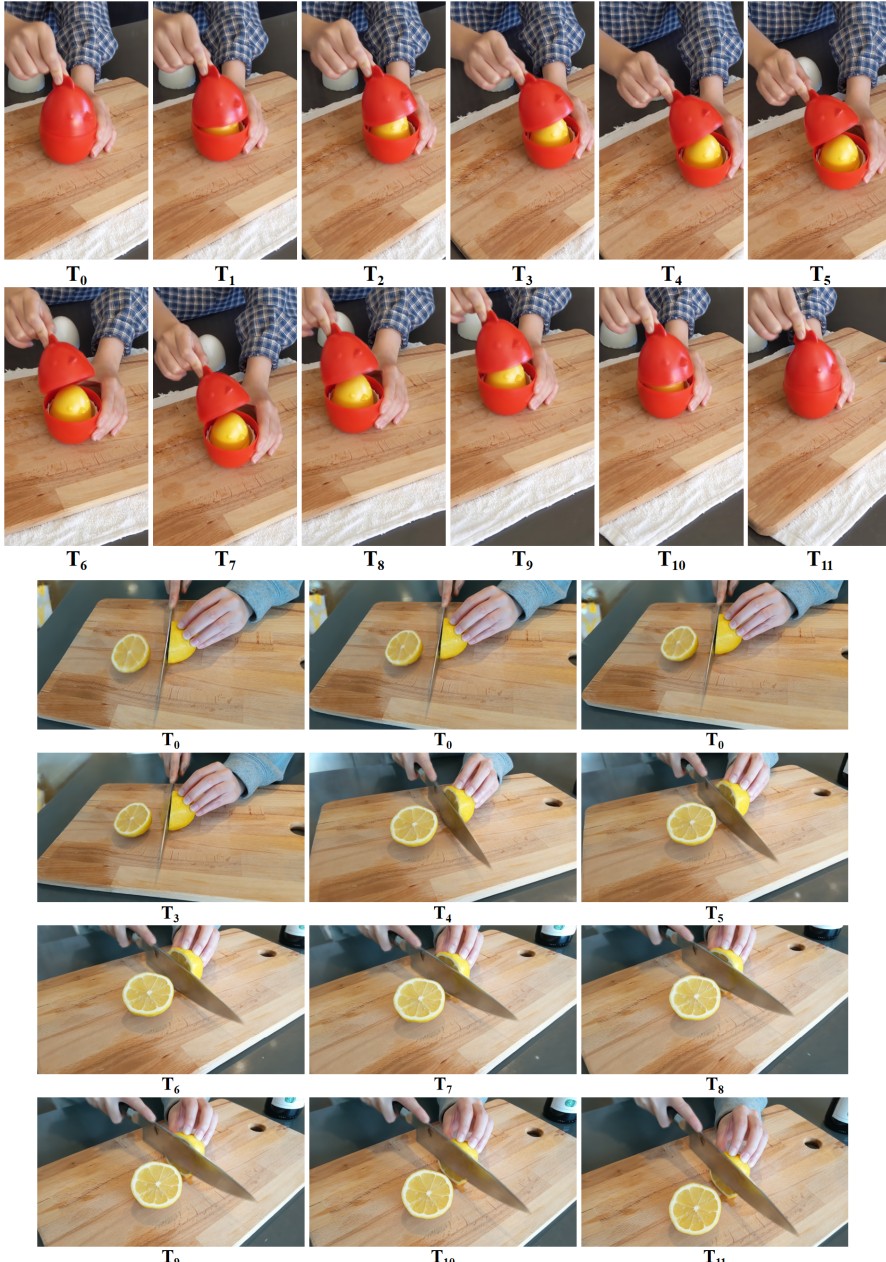

Figure 7: Sequence of 12 consecutive inference results from Mango-GS ($T = 6$). Each group displays a complete inference output at specific timestamps.

