# OpenReview forum: "Mango-GS: Enhancing Spatio-Temporal Consistency in Dynamic Scenes Reconstruction using Multi-Frame Node-Guided 4D Gaussian Splatting"
_ICLR.cc/2026/Conference — ICLR 2026 Poster_

### Official Review · Reviewer_NRrk · 2025-10-27

**Soundness:** 4
**Presentation:** 3
**Contribution:** 4
**Rating:** 6
**Confidence:** 4

**Summary:**

The paper tackles real-time dynamic scene reconstruction from monocular videos based on SC-GS, aiming to model long-range, non-linear motion with temporal coherence. The authors propose Mango-GS, which couples 3D Gaussian Splatting with a node-guided multi-frame network:  (1) an MLP backbone interleaves temporal self-attention and gated fusion to propagate dependencies across a window of $T$ frames and predict per-node deformations; (2) training employs temporal input masking plus a composite loss with frame loss and motion loss. Experiments on Neural 3D Video, and HyperNeRF-vrig report consistent rendering quality improvement and stability while remaining real-time rendering speed.

**Strengths:**

- This paper is well-written and easy to understand.
- I appreciate the paper’s attempt to process multiple frames jointly. Handling T frames simultaneously and introducing self-attention across different timestamps make sense, and this idea could inspire future work on monocular (multi-view) dynamic reconstruction.”
- The ablation is appreciated to show the important contribution of each part.
- Across different real-world datasets, the results are consistently strong, with visualizations aligning well with the quantitative metrics.

**Weaknesses:**

- Missing references for some important dynamic reconstruction works:
  - [NeurIPS 2024] Grid4D: 4D Decomposed Hash Encoding for High-Fidelity Dynamic Gaussian Splatting, by Jiawei Xu et al.
  - [NeurIPS 2024] Splatter a Video: Video Gaussian Representation for Versatile Processing, by Yang-Tian Sun et al.
- Using a lightweight MLP to compute the implicit distance between control points and Gaussians is insightful. However, I don’t find the benefit of this design immediately clear from Figure 2. Have the authors explored alternative formulations, such as a codebook approach similar to VQ-VAE?
- A minor suggestion: Figure 3 does not clearly convey self-attention across time. Also, if my understanding is correct, position and timestamp should be provided jointly as input to the MLP?
- Minor typo errors:
  - L23: `which provides` -> `which provide`
  - L24: `prevents correspondence error` -> `prevents correspondence errors`
  - L695-696: `Each groups displays` -> `Each group displays`

**Questions:**

1. What is the approximate training time? How much overhead is there relative to SC-GS?
2. Could you clarify how the temporal input masking is constructed (random or structured across time)? What masking ratio do you use, and is it fixed or scheduled during training?

**Details Of Ethics Concerns:**

None.

---

> ### Author Response · Authors · 2025-11-24
>
> We thank you for the valuable references and detailed questions regarding our implementation choices. We now updated the related work with the suggested citations, clarified the MLP input structure and masking strategy, and provided a specific training cost analysis.
>
> **W1: Additional Citations.** We have incorporated citations for Grid4D and Splatter a Video in the related work section. We added a discussion highlighting how Mango-GS differs from them, specifically contrasting our node-guided multi-frame motion modeling with their decomposed 4D hash encoding and video-oriented representations.
>
> **W2: Different Node-Gaussians Connections.** We have experimented with a codebook-style alternative for modeling the node–Gaussian affinity, and found that it can give slightly better reconstruction metrics on scenes with larger static background, but at the cost of a noticeably larger model. Since our goal in this work is to keep the representation lightweight and easy to deploy, we chose the simple MLP-based metric for Mango-GS and leave a more systematic exploration of codebook-based formulations as promising future work.
>
> **W3: Input Clarification.** We apologize for the confusion that have caused. We confirm that positions and multi-frame timestamps are provided jointly as input to the MLP.
>
> **W4: Typo Corrections.** We thank the reviewer for the careful reading; all identified typos have been corrected in the revised manuscript.
>
> **Q1: Training Cost Analysis.** We train for the same 40k iterations as SC-GS. On a single RTX 3090 (HyperNeRF-vrig dataset), Mango-GS takes ≈68 minutes compared to 55 minutes for SC-GS. We consider this modest overhead (≈24%) acceptable given the significant gains in reconstruction quality (23.6 v.s. 26.2 in PSNR) and rendering FPS (24.5 v.s. 149.5 in FPS).
>
> **Q2: Masking Strategy Details.** We use a simple random masking across time: for each training window we uniformly sample a fixed proportion of timestamps and drop their time embeddings (the same temporal mask is set to zero to keep alignment). The masking ratio is kept constant at 0.2 throughout training (no schedule), and the mask is resampled every iteration so the model sees different masked patterns over time rather than a fixed structured pattern.

---

### Official Review · Reviewer_ytGT · 2025-10-29

**Soundness:** 3
**Presentation:** 3
**Contribution:** 3
**Rating:** 6
**Confidence:** 4

**Summary:**

The paper first points out that existing Gaussian Splatting methods rely on per-frame optimization, causing them to memorize frame-specific states rather than learning true motion dynamics. To address this, it proposes Mango-GS, a multi-frame, node-guided 4D Gaussian Splatting framework that introduces a decoupled control node representation and a temporal attention network. The method aims to enable efficient, temporally consistent, and high-fidelity dynamic scene reconstruction.

**Strengths:**

* The paper clearly identifies a core weakness in existing dynamic 3D Gaussian Splatting methods, namely their reliance on per-frame optimization, which causes temporal inconsistency and overfitting to instantaneous states. The motivation for introducing a multi-frame modeling framework is well justified and directly addresses this limitation.

* The proposed multi-frame temporal deformation network interleaves MLP layers with temporal self-attention blocks and a gated fusion mechanism. This hybrid design effectively captures long-range motion dependencies while remaining computationally efficient. It represents a well-balanced approach between expressive modeling and real-time rendering.

* The paper is well organized, with clear reasoning and smooth transitions between sections, making the overall presentation easy to follow.

**Weaknesses:**

* My main concern regarding the design of Mango-GS lies in the insufficient justification for using a Transformer to address temporal inconsistency. First, the advantage of Transformer is its ability to capture long-range dependencies, but Table 2 shows that the optimal temporal window is only six frames. Is using a Transformer to model such a short sequence truly necessary, and does the computational cost justify the potential performance gain? Furthermore, several prior works have also explored multi-frame motion modeling to improve temporal coherence. The paper should include an ablation study comparing the Transformer with other mechanisms that leverage multi-frame motion information, in order to clearly demonstrate the advantages of using a Transformer in this context.

* In Figure 4, the highlighted regions should be enlarged for better visual inspection.

* In Figure 5, the visual differences among compared methods are not very clear. For example, in the *Peel-Banana* scene, the results of 4DGS and Mango-GS appear quite similar, and in the *Cut-lemon* scene, differences among all methods are subtle. The paper should select frames with more noticeable differences or to use highlighted and zoomed-in regions to better demonstrate the superiority of Mango-GS. Based on the current visualizations, I do not observe a significant advantage of the proposed method.

* In the experimental section, all compared methods are from 2023-2024. The paper should include comparisons with the latest approaches and should report detailed per-scene quantitative results.

* The paper should also evaluate the method on more challenging datasets, such as the iPhone dataset, to further validate its robustness and generalization ability.
* The discussion of related work should be expanded to include more recent studies, as the current review is not sufficiently comprehensive.
* If possible, I hope to see video results to more intuitively understand the advantage of Mango-GS in motion consistency.

**Questions:**

See weaknesses.

---

> ### Author Response · Authors · 2025-11-24
>
> We appreciate your constructive feedbacks. In the revised version, we have justified the choice of the Transformer module with ablation studies, expanded comparisons to include recent methods (MotionGS and TimeFormer suggested by reviewer ZZyk), provided additional analysis on iPhone datasets, and improved visualizations in both the main paper (zoomed-in figures) and supplementary material (video results).
>
> **W1: Justification for Temporal Transformer.** Although the temporal window is relatively short (T=6), our temporal attention with Transformer is applied per control node, which must jointly reason about heterogeneous frame rates, occlusions, and node-specific motion patterns. Our experiments show that simple MLPs as the temporal attention struggle with these variations (Tab. 3, Abl. 4&5, 27.30 vs. 27.78 PSNR) and yields a ~12% worse temporal-LPIPS (a temporal metric suggested by reviewer 6RAS). Given the low computational overhead in the sparse node space, the Transformer offers the best trade-off between cost and spatio-temporal quality.
>
> **W2 & W3: Visual Clarity Improvements.** We have enlarged the highlighted regions in Fig. 4 and added zoomed-in crops in Fig. 5. These updates make the improvements more apparent: specifically, Mango-GS preserves finer surface details on the Peel-Banana scene and reconstructs text/edges more clearly on the knife in the Cut-lemon scene compared to baselines.
>
> **W4: Additional Comparisons.** We have added comparisons with recent methods, including MotionGS and TimeFormer (2025), with detailed per-scene quantitative results in the **Supplementary Material Table 7**. Mango-GS matches or surpasses these methods on reconstruction metrics while achieving significantly higher FPS due to our efficient node-guided design.
>
> **W5: Evaluation on iPhone Dataset.** We conducted preliminary experiments on an iPhone dataset and observed that Mango-GS broadly follows the same trend as other datasets although the absolute numbers are limited by challenges such as monocular capture, noisy camera poses, and motion blur, which require careful dataset specific tuning and preprocessing. Due to space and time constraints, we therefore focus the main paper on widely used benchmarks with standardized protocols, and leave a more systematic and better tuned study for future work. However, we now present results of evaluating Mango-GS on four commonly used iPhone scenes (Teddy, Windmill, Spin, and Cat) and report the quantitative comparisons. On these sequences, Mango-GS achieves comparable or better PSNR, SSIM, and tLPIPS than 4DGS, SC-GS, and MotionGS, while maintaining real-time rendering speed.
>
> | Method | PSNR$\uparrow$ | MS-SSIM$\uparrow$ | tLPIPS$\downarrow$ | FPS $\uparrow$ |
> | :--- | :---: | :---: | :---: | :---: |
> | 4DGS | **18.3** | 0.41 | 0.138 | 64 |
> | SC-GS | 15.8 | 0.28 | 0.172 | 19.7 |
> | MotionGS | 17.3 | 0.35 | 0.149 | 35.2 |
> | **Mango-GS (ours)** | 18.1 | **0.42** | **0.129** | **116** |
>
> **W6: Related Work.** We have expanded the related work section to include a broader discussion of recent dynamic Gaussian splatting and temporal modeling approaches.
>
> **W7: Video Demonstrations.** We have included video results in the Supplementary Material, which provide a dynamic view of Mango-GS's actual performance and temporal stability.

---

### Official Review · Reviewer_ZZyk · 2025-10-30

**Soundness:** 2
**Presentation:** 2
**Contribution:** 2
**Rating:** 4
**Confidence:** 3

**Summary:**

This paper presents a 4D scene reconstruction approach that ensures both photorealistic detail and temporal coherence. It leverages a temporal transformer to model complex motion dependencies across multiple frames, guided by a set of sparse control nodes with decoupled position and latent codes. This design enables stable motion representation and prevents correspondence errors during large movements. Trained end-to-end with multi-frame objectives, Mango-GS achieves state-of-the-art reconstruction quality and real-time rendering performance.

**Strengths:**

1. Decoupling the 4DGS makes sense.
2. This paper presents a good FPS performance.

**Weaknesses:**

1. For the quantitative comparison in Table 1. The improvement in PSNR is limited.
2. The visualized comparison is weak. It is better to provide video comparison in supplementarials.

**Questions:**

1. Please include the cites in Table 1.
2. Please add the discussions with other approaches that focus on the temporal modeling of Gaussians. e.g. A, B.

[A] MotionGS: Exploring Explicit Motion Guidance for Deformable 3D Gaussian Splatting
[B] TimeFormer: Capturing Temporal Relationships of Deformable 3D Gaussians for Robust Reconstruction

---

> ### Author Response · Authors · 2025-11-24
>
> We thank you for the helpful suggestions regarding baselines and visualizations. In the revised version, we have added the recommended citations, expanded the comparison to highlight our efficiency advantages, and provided video results in the supplementary material.
>
> **W1: Efficiency and Temporal Coherence.** Beyond PSNR gains, Mango-GS demonstrates a significant advantage in real-time capability, reaching 149.5 FPS compared to 45.2 FPS for the second-best baseline (E-D3DGS) . Furthermore, our method achieves superior temporal coherence, evidenced by lower temporal metrics: t-LPIPS (0.0196 vs. 0.0229 (MotionGS, which is the second best method)), which is suggested by reviewer 6RAS, indicating better stability than competing methods even when PSNR is comparable.
>
> **W2: Video Results.** We have included video results in the Supplementary Material. These visual comparisons more clearly showcase the actual performance and temporal stability of Mango-GS in dynamic scenes.
>
> **Q1 & Q2: Temporal Modeling Methods.** We have added the suggested citations to Table 1 and extended the discussion in the Related Works section. Our revised analysis emphasizes that Mango-GS not only matches or surpasses these methods on reconstruction metrics but also achieves significantly higher FPS due to our multi-frame design.

---

### Official Review · Reviewer_6RAS · 2025-10-31

**Soundness:** 3
**Presentation:** 2
**Contribution:** 3
**Rating:** 6
**Confidence:** 4

**Summary:**

This paper presents Mango-GS, a multi-frame node-guided framework for dynamic 3D scene reconstruction using Gaussian splatting. The key innovation lies in decoupling control nodes into canonical positions and learnable feature codes, combined with a temporal transformer that processes multiple frames simultaneously to learn motion dynamics. The sparse control nodes guide dense 3D Gaussians through learned k-NN relationships, enabling efficient temporal modeling while avoiding per-frame overfitting. Experiments on HyperNeRF and Neural 3D Video datasets demonstrate state-of-the-art reconstruction quality with real-time rendering speeds of 149.5 FPS and reasonable storage requirements of 60 MB.

**Strengths:**

* The decoupled node representation is a well-motivated design that elegantly addresses the neighborhood drift problem in large motion scenarios.
* The multi-frame temporal attention mechanism represents a significant departure from per-frame optimization strategies prevalent in prior work. This design enables the model to learn motion patterns rather than memorize instantaneous states, leading to improved temporal coherence as evidenced by both quantitative metrics and qualitative visualizations.
* The ablation studies systematically validate each component's contribution, and the method achieves excellent rendering speed while maintaining competitive storage efficiency.

**Weaknesses:**

* The theoretical justification for why the decoupled representation prevents neighborhood drift is primarily empirical. While Figure 2 provides visual evidence, a more rigorous analysis of the learned feature space and how it maintains semantic consistency under large deformations would strengthen the claims.
*  The motion-aware loss components ($L_{diff}, L_{dir}$) are mentioned but never formally defined in the main paper.
* The evaluation focuses heavily on PSNR/SSIM metrics, but temporal consistency evaluation is limited. While motion-aware loss is used during training, there are no temporal metrics in the evaluation (e.g., temporal LPIPS, or frame-to-frame stability measures). This is a significant omission for a method claiming temporal coherence as a primary contribution.
* The comparison with SC-GS shows lower performance (Table 1: 30.20 vs 31.89 PSNR), but SC-GS also uses control nodes. The paper doesn't adequately explain why their approach differs beyond adding temporal attention. A more detailed comparison highlighting the specific differences in node design and their impact would be valuable.
* The multi-frame sampling strategy uses an arbitrary 7:3 ratio between sparse stride and interpolation samplers with no justification or ablation. How sensitive is training to this ratio? Why not 5:5 or 9:1?

**Questions:**

* How does the method handle scenarios where objects enter or leave the scene mid-sequence? Does the k-NN relationship remain stable, or do you need special handling for appearance/disappearance events? Please provide examples or discuss this limitation.
* You claim T=6 provides the best balance (Table 2a), but the performance difference between T=6 and T=8 is marginal (28.35 vs 28.24 PSNR) while T=8 achieves higher FPS (156.2). Can you provide more analysis on this trade-off? Are there specific motion types that benefit from larger T?
* The method uses 2048 initial control nodes which are dynamically densified/pruned. What are the densification and pruning criteria?
* In Figure 5, your method shows sharper results than baselines. However, could you provide temporal consistency metrics? Static frame comparisons don't fully validate the claimed temporal coherence improvements. Have you conducted user studies comparing temporal quality?
* The top-k hard-frame loss selects the k frames with highest error per batch. What is k as a fraction of batch size? If k is too small, does this ignore most training signals? If k is too large, does this reduce to average loss? In Table 3, adding top-k loss improves LPIPS from 0.084 to 0.077, but this could simply be due to focusing on high-frequency details in difficult frames rather than improving overall consistency.

---

> ### Author Response · Authors · 2025-11-24
>
> We sincerely thank you for the insightful comments and constructive suggestions. We have carefully revised the manuscript to address your concerns. Key updates include: explicit definitions of motion-aware loss terms, additional temporal consistency metrics (tLPIPS), new object-level semantic stability studies, and ablation studies on data ratios.
>
> **W1: Object-level semantic stability.** To provide a rigorous analysis, we conducted an object-level semantic stability study. We identified control nodes associated with specific object masks and measured the fraction of influenced Gaussians remaining inside the mask over time. As detailed in **Supplementary Sec. A.3**, Mango-GS consistently achieves a higher stability ratio than the coupled baseline, proving that our nodes remain stably attached to semantic regions.
>
> **W2: Formal definitions.** We have updated the main paper to explicitly provide formal definitions for each motion-aware loss term to ensure clarity.
>
> **W3: Temporal consistency metrics.** We have included the temporal perceptual metric tLPIPS in the revision. The additional experiments demonstrate that our method consistently achieves lower tLPIPS scores compared to baselines, quantitatively supporting our claim of improved temporal coherence.
>
> **W4: Comparison with SC-GS.** While both methods use control nodes, the design differs significantly. SC-GS ties control nodes directly to Gaussian primitives, whereas we explicitly decouple the node’s semantic code from its spatial position. This design allows our k-NN graph to be built in a semantically enriched space, ensuring stability even under large deformations.
>
> **W5: Data ratio ablation.** We added an ablation study on the 7:3 ratio in **Supplementary Tab. 4**. Results indicate that performance is relatively stable between ratios of 9:1 and 5:5, with our chosen 7:3 setting providing the optimal balance.
>
> **Q1: Handling appearing/disappearing objects.** Our method does not need special handling for objects that appear or disappear. For example, in the HyperNeRF Chicken scene, the small yellow chick under the lid only becomes visible halfway through, and its Gaussians already exist beforehand but are fully occluded by the lid, so they simply do not contribute to rendering until the lid moves. In general, newly visible regions are captured by our standard densification (their Gaussians start receiving gradients and get linked to nearby nodes), while long-unobserved regions are naturally hide into the background. Besides, our multi-frame, motion-aware modeling then helps these regions follow coherent motion, improving both accuracy and temporal stability without extra heuristics.
>
> **Q2: Choice of T=6 over T=8.** We treat PSNR as the primary metric in our main table, so we report (T=6) as the default. While T=8 yields higher FPS, we observed a slight degradation in tLPIPS (0.0196 vs 0.0197 on HyperNeRF-vrig) and increased training costs. We selected T=6 as the best balance between reconstruction quality, temporal stability, and efficiency.
>
> **Q3: Node densification and pruning.** Nodes accumulating large reconstruction errors are densified to capture detail, while nodes rarely selected as neighbors are pruned. By aggregating these statistics over multi-frame batches, our method naturally adapts topology: adding nodes in regions with complex motion and removing redundancy in static areas.
>
> **Q4: Temporal quality evaluation.** As mentioned in W3, we have added tLPIPS results (Table 1) which show our method outperforms baselines in temporal consistency. While we agree a user study is valuable, we prioritized quantitative metrics for this revision and leave extensive perceptual user studies for future work.
>
> **Q5: Top-k strategy.** We set k to a fixed ratio (0.6× batch size) and recompute Top-k frames at every iteration. This ensures gradients are reweighted toward currently hard frames. Additionally, overall consistency is enforced by the motion-aware loss operating on all frame differences.

---

### Author Response · Authors · 2025-11-27

Dear Reviewers,

I hope this message finds you well. As the discussion period is nearing its end, we would like to briefly summarize the main revisions and clarify that we are very happy to address any remaining concerns. Your feedback has been extremely helpful in improving the paper.

**Brief recap of the Mango-GS**

Mango-GS is a multi-frame, node-guided 4D Gaussian splatting framework that aims to jointly achieve **high reconstruction quality and high rendering FPS**. It combines:
* A **temporal Transformer** over a window of frames to model per-node motion trajectories;
* **Decoupled control nodes** (position + latent code) that serve as stable semantic anchors for motion;
* An efficient **k-NN node–Gaussian coupling**, so dense Gaussians inherit these coherent motions while maintaining high FPS.

**Key updates during rebuttal**

To better support our claims and address the reviewers’ comments, we made the following main additions:

1. Improved qualitative results and videos
* Updated figures with **larger zoom-in regions** and added an illustration of node–Gaussian semantic groups.
* Provided **video results** in the supplementary material, making the temporal advantages of Mango-GS clearer.

2. Stronger baselines and datasets
* Included new temporal baselines **MotionGS** and **TimeFormer**, and added missing recent works (e.g., Grid4D, Splatter a Video) to the related work and comparisons.
* Reported additional results on the **iPhone dataset**, where Mango-GS remains competitive or better while preserving real-time speed.
* Added a **per-scene** breakdown on HyperNeRF-vrig to show that improvements are consistent across scenes.

3. Temporal coherence: metrics, loss, and analysis
* Added **temporal LPIPS (tLPIPS)** as a core evaluation metric in the main comparison and ablations; Mango-GS achieves the best tLPIPS while keeping very high FPS.
* Introduced an **object-level node–Gaussian semantic consistency study** and visualized correspond k-NN structure, showing that decoupled nodes with learned affinity stay attached to semantic regions over time significantly better than the coupled baseline.
* Formally defined the motion-aware loss operating on frame differences and clarified all terms.

4. Architectural and hyperparameter ablations
* Compared the **temporal Transformer vs. temporal MLP**, and added **multi-frame dataset sampling ratio ablation**. These studies show that the chosen configuration gives the best balance between reconstruction quality, temporal stability (tLPIPS), and efficiency.

If there are any additional points you would like us to clarify or if something still feels undersupported, please let us know—we would be very happy to respond promptly and further refine the paper where possible.

Thank you again for your time, effort, and thoughtful feedback in reviewing our work.

Best regards,

**Authors of “Mango-GS”**

---

### Author Response · Authors · 2025-12-03
**Summary of Mango-GS**

### Dear Area Chair,

Thank you very much for your time in handling our submission Mango-GS under the new procedure.

In this note, we briefly summarize our method and the main experiments. For the detailed, point-by-point rebuttal to each reviewer and the exact changes made in the revised paper, we kindly refer you to our previous comments.

---

### Mango-GS provides temporally coherent multi-frame 4D Gaussian splatting

Existing dynamic 3D Gaussian splatting methods are largely **per-frame**, which tends to memorize instantaneous states and leads to temporal flicker and inconsistent motion across frames. Mango-GS targets this by a multi-frame 4D Gaussian splatting framework that:

1. **Temporal multi-frame modeling.** Processes a short window of frames (T=6) jointly with a temporal Transformer to aggregate multi-frame evidence and capture complex motion patterns with modest overhead.

2. **Node-guided semantic motion.** Uses decoupled control nodes (canonical 3D position + latent code) together with a learned node–Gaussian affinity, so motion is propagated in a semantic neighborhood rather than purely in Euclidean space, reducing neighborhood drift and improving object-level temporal coherence.

3. **Training strategy for stable trajectories.** Combines temporal input masking, a multi-frame sampling strategy (sparse-stride and interpolation windows when constructing training sequences), and a composite loss (top-k hard-frame photometric loss plus a motion-aware loss on frame differences) to encourage stable trajectories instead of per-frame overfitting.

---

### Experimental comparisons demonstrate strong accuracy and efficiency

We evaluate Mango-GS on HyperNeRF-vrig and Neural 3D Video, comparing against 4DGS, SC-GS, D-3DGS, E-D3DGS, GaGS, and recent temporal methods MotionGS and TimeFormer. Metrics include PSNR, MS-SSIM, LPIPS, temporal LPIPS (tLPIPS), as well as FPS and storage size.

On HyperNeRF-vrig, Mango-GS:

1. **Achieves state-of-the-art reconstruction quality**, improving PSNR over E-D3DGS from about 25.4 to 26.2 while maintaining competitive MS-SSIM and LPIPS (main paper).

2. **Achieves the best temporal coherence**, with tLPIPS ≈ 0.0196 vs. 0.0229 for MotionGS (the second-best temporal method) (main paper).

3. **Reaches ≈149.5 FPS**, significantly faster than the next best baseline (E-D3DGS ≈ 45.2 FPS), with compact storage (~60 MB) (main paper).

On Neural 3D Video, Mango-GS likewise matches or surpasses competing methods in PSNR/SSIM and tLPIPS while remaining real-time, confirming that the temporal multi-frame design does not sacrifice efficiency (main paper).

---

### Ablation studies justify key temporal, node, and training choices

To support the design decisions, we conduct ablations along three axes: temporal modeling, node design, and training strategy.

1. **Temporal module and window length.** Replacing the temporal Transformer with a simple temporal MLP reduces PSNR and leads to ≈12% worse tLPIPS, showing that attention is beneficial even for a short window (T=6). Varying T shows that T=6 offers the best compromise between PSNR, tLPIPS, and FPS, while larger T brings diminishing returns with higher training cost (main paper).

2. **Neighborhood structure.** Varying the node–Gaussian neighbor count K shows that K=3 gives the lowest tLPIPS, whereas larger K over-smooths motion and degrades temporal coherence (main paper).

3. **Multi-frame sampling and loss weighting.** We study different ratios between sparse-stride and interpolation sampling windows (e.g., 9:1, 7:3, 5:5). Performance is stable between 9:1 and 5:5, with 7:3 producing the best joint PSNR / MS-SSIM / tLPIPS balance. Besides, the top-k hard-frame loss (k ≈ 0.6 × batch size), combined with the motion-aware loss over all frame differences, further improves LPIPS/tLPIPS by focusing supervision on difficult frames without losing coverage of easier ones (supplementary materials).

---

---

> ### Author Response · Authors · 2025-12-03
>
> ### Validation experiments support temporal coherence and semantic stability
>
> Beyond standard metrics, we conduct validation experiments to directly examine temporal behavior and semantic consistency:
>
> 1. **Object-level semantic stability study**. We track object masks over time, identify nodes associated with each object, and measure the fraction of influenced Gaussians that remain inside the object region across frames. Mango-GS achieves a substantially higher stability score than a coupled baseline, confirming that the node-guided design reduces neighborhood drift and keeps node–Gaussian groups semantically coherent (supplementary materials).
>
> 2. **Qualitative visualizations and videos**. The results show that Mango-GS produces smoother motion and fewer flickering artifacts than baselines, complementing the tLPIPS results (main paper and supplementary materials).
>
> 3. **Preliminary tests on iPhone dataset** (Teddy, Windmill, Spin, Cat). These experiments indicate that Mango-GS remains comparable or better than 4DGS, SC-GS, and MotionGS in PSNR / SSIM / tLPIPS while still running at >100 FPS. Given the challenges of noisy poses and motion blur, we treat this as an initial robustness check and keep our main conclusions based on standardized benchmarks (rebuttal).
>
> ---
>
> We are sincerely grateful for your time and effort in handling our submission under the new procedure.
> We hope this brief note provides a clear overview of Mango-GS and its empirical validation, and we appreciate your consideration.
>
> Best regards,
>
> **Authors of “Mango-GS”**

---

### Meta-Review · Area_Chair_wfe4 · 2026-01-04

**Summary:**

This paper proposes Mango-GS, a multi-frame, node-guided 4D Gaussian splatting framework for dynamic scene reconstruction, aiming to improve spatio-temporal consistency while maintaining real-time rendering performance. The main strength of the work lies in its principled move beyond per-frame optimization by jointly modeling short temporal windows and propagating motion through decoupled control nodes, which provides a stable mechanism for handling complex and large motions. Reviewers raised concerns primarily about the strength of the empirical evidence for temporal coherence, the clarity and justification of several design choices, and the completeness of comparisons and visualizations. The authors have addressed most of these concerns during the rebuttal phase. Taking the reviews and rebuttal together, I recommend accepting this paper.

**Reviewer Concerns:**

The main reviewer concerns on temporal evaluation, missing comparisons, architectural justification, and qualitative results were largely addressed. Remaining issues, like modest metric gains or absence of user studies, are minor and do not weaken the core contributions.

**Reviewer Scores:**

Reviewers who raised technical or empirical concerns would likely have maintained or slightly increased their scores given the additional experiments and clarifications provided during rebuttal.

---

### Decision · Program_Chairs · 2026-01-26

Accept (Poster)